# Synthesis and Characterization of Sn/SnO_2_/C Nano-Composite Structure: High-Performance Negative Electrode for Lithium-Ion Batteries

**DOI:** 10.3390/ma15072475

**Published:** 2022-03-27

**Authors:** Jaffer Saddique, Honglie Shen, Jiawei Ge, Xiaomin Huo, Nasir Rahman, Muhammad Mushtaq, Khaled Althubeiti, Hamza Al-Shehri

**Affiliations:** 1Jiangsu Key Laboratory of Materials and Technology for Energy Conversion, College of Materials Science and Technology, Nanjing University of Aeronautics and Astronautics, Nanjing 210016, China; jafferphy@gmail.com (J.S.); 18551755923@sina.cn (J.G.); huoxm0928@163.com (X.H.); 2Department of Physics, University of Lakki Marwat, Lakki Marwat 28420, Khyber Pakhtunkhwa, Pakistan; nasir@ulm.edu.pk; 3Key Laboratory of Advanced Functional Materials of Education Ministry of China, College of Materials Science and Engineering, Beijing University of Technology, Beijing 100124, China; mushtaqphy009@yahoo.com; 4Department of Chemistry, College of Science, Taif University, P.O. Box 11099, Taif 21944, Saudi Arabia; k.athubeiti@tu.edu.sa; 5Chemistry Division, King Khaled Military Academy, SANG, Riyadh 11495, Saudi Arabia; h.s.alshehri@outlook.com

**Keywords:** Sn/SnO_2_/C composite anode material, lithium-ion battery (LIBs), energy storage, synthesis, electrochemical performance

## Abstract

Tin oxide (SnO_2_) and tin-based composites along with carbon have attracted significant interest as negative electrodes for lithium-ion batteries (LIBs). However, tin-based composite electrodes have some critical drawbacks, such as high volume expansion, low capacity at high current density due to low ionic conductivity, and poor cycle stability. Moreover, complex preparation methods and high-cost carbon coating procedures are considered main challenges in the commercialization of tin-based electrodes for LIBs. In this study, we prepared a Sn/SnO_2_/C nano-composite structure by employing a low-cost hydrothermal method, where Sn nanoparticles were oxidized in glucose and carboxymethyl cellulose CMC was introduced into the solution. Scanning electron microscope (SEM) and transmission electron microscope revealed the irregular structure of Sn nanoparticles and SnO_2_ phases in the conductive carbon matrix. The as-prepared Sn/SnO_2_/C nano-composite showed high first-cycle reversible discharge capacity (2248 mAhg^−1^) at 100 mAg^−1^ with a first coulombic efficiency of 70%, and also displayed 474.64 mAhg^−1^ at the relatively high current density of about 500 mAg^−1^ after 100 cycles. A low-cost Sn/SnO_2_/C nano-composite with significant electrochemical performance could be the next generation of high-performance negative electrodes for LIBs.

## 1. Introduction

Rechargeable lithium-ion batteries (LIBs) have been used during the last few decades as the main power source of choice with tremendous applications in many fields, such as portable optoelectronics devices, mobile phones, laptops and cameras, etc. [1,2,3,4]. However, given the high demands and rapid development of electric vehicles, lithium-ion batteries are now urgently required, especially those with high energy density, a long cycle life, and fast charging capacity [5,6,7]. A lot of expectations are on LIBs in terms of stable performance to meet the high demands of the current electronic market. However, the conventional anode material graphite (372 mAhg^−1^) used commercially is incapable of fulfilling the requirement of the current demands of power due to its low theoretical capacity [8,9,10]. According to the current demands, it a priority to explore new-generation electrode materials with excellent electrochemical performances, including high lithiation capacity, long cycle lifespan, low cost, and that is eco-friendly, to boost up the overall performances of LIBs. In this context, much work has been done to introduce high-performance electrode materials for LIBs. Among these, metallic tin is a non-toxic, environmentally friendly, low-cost and highly abundantly available metal with outstanding specific theoretical capacity (994 mAhg^−1^) for LIBs [11,12]. However, metallic tin electrodes have some serious drawbacks concerning the dramatic volume expansion >300% during the lithiation/delithiation process resulting in capacity fading, rapid capacity decay, and low coulombic efficiency, which always limits their potential applications [13,14]. Tin oxide SnO_x_, specifically SnO_2_ and SnO, have attracted attention as a potential next-generation anode owing to their high theoretical capacity of 1494 mAhg^−1^ and 1273 mAhg^−1^, respectively [15,16]. In the literature, different strategies have been applied to obtain different structures of tin oxide in order to improve the electrochemical performance [17,18]. By controlling and manipulating important parameters of the SnO_2_, such as the size of the electrode nanostructure and the confinement of the active material in a carbonaceous matrix to prevent the agglomeration of the nanostructures upon cycling, it is possible to increase the amount of lithium-ion reversibility during conversion reactions [19,20,21]. During lithiation, the reaction mechanism of SnO_2_ can be described as two stages: the (1) conversion reaction and (2) alloying reaction, which are given as follows:

In the conversion stage, SnO_2_ is converted to form Li_2_O and the elemental Sn as expressed by the following equation:SnO_2_ + 4Li^+^ + 4e^−^ ↔ 2Li_2_O + Sn(1)
which contributes to a capacity of 711 mAhg^−1^.

In the second stage, the alloying reaction taking place can be expressed as the following equation:Sn + xLi^+^ + 4e^−^ ↔ LixSn (0 < x < 4.4)(2)
to which a further 783 mAhg^−1^ can be added over its theoretical capacity, and it can reach as high as 1494 mAhg^−1^ [22,23]. Besides these, SnO_2_ has some drawbacks when used as a negative electrode for LIBs, such as high volume expansion during lithiation, poor conductivity, and large first-cycle capacity, which lead to electrode pulverization and electrical exfoliation of the active material from the current collector and further hinder their application, resulting in poor cycle performance, rapid capacity decay, and low coulombic efficiency [24,25,26]. Different morphology and nanostructures, and the introduction of Sn-based anode into the carbon matrix are helpful to overcome all these obstacles while utilizing and introducing a negative electrode material for LIBs with such high theoretical capacity [27,28]. Herein, an Sn/SnO_2_/C nano-composite anode, prepared by facile hydrothermal method, is expected to be able to enhance the performance of LIBs, especially in its real capacity with stable cyclability and coloumbic efficiency. Structural characterizations confirm the existence of Sn and SnO_2_ in the carbon conductive framework, ensuring Sn/SnO_2_/C-based LIBs that possess considerable enhancement in terms of cycle stability as well as other enhanced performances.

## 2. Experimental

### 2.1. Preparation of Sn/SnO_2_/C in Carbon Matrix

A typical hydrothermal method was used for the preparation of Sn/SnO_2_/C nano-composite materials. In detail, the desired molar ratio of tin chloride dehydrate SnCl_2_·2H_2_O (99.9%, Aladdin) and Glucose C_6_H_12_O_6_ (99%, Aladdin Nanjing Chemical Reagent Co., LTO) of about 0.02 mole (4.51 g) and 0.02 mole (3.6 g), respectively, were mixed in 50 mL deionized water. The solution was stirred for 15 min and then 4 g (weight %) carboxymethyl cellulose CMC was added slowly and gradually into the solution and continually stirred for more than 2 h in order to get a homogenous solution milky in color. All the reagents were of the analytical grade and used without any further purification. The solution was then transferred to a 100 mL Teflon-lined autoclave and heated to 200 °C and kept at this temperature for 24 h to get the SnO_x_ nanoparticles. The solution was cooled naturally to room temperature and was then centrifuged at 9000 r/min several times. Then, the solution was washed thoroughly by using de-ionized water and ethanol to obtain more pure products. Finally, the products were transferred to a vacuum oven and kept at a temperature of 90 °C overnight to dry them. To get the Sn/SnO_2_/C, the resultant products were ground with the help of a pestle and mortar and then annealed at 800 °C at the rate of 5 °C/min for 4 h under argon atmosphere. The used carboxymethyl cellulose CMC polymer was converted into carbon after high-temperature treatment by following the annealing process. The experimental steps were exercised repeatedly to make it more accurate.

### 2.2. Materials Characterization

The crystalline structure and phase confirmation of the as-synthesized Sn/SnO_2_/C material was characterized via room temperature powder X-ray diffractometer (XRD) in the range of 10–80° on Broker D8 with Cu Kalpha radiation. Scanning Electron Microscopy (SEM model FEI, Quanta 650, HITACHI, Japan) was carried out to analyze the morphological structure of the as-synthesized material. Energy dispersive spectroscopy (EDS) along with SEM was performed to confirm the elemental distribution mapping of the elements contained by the prepared materials. Transmission electron microscopy (TEM) along with selected area electron diffraction (SAED) and high-resolution transmission microscopy (HRTEM, Tokyo, Japan) was performed on EFI Tecnai G2s-Twin instrument with electron gun operating at 200 KV to further confirm the atomic distribution and size of the synthesized Sn/SnO_2_/C material. X-ray photoelectron spectroscopy (XPS, PHI-5000 VersaProbe Ulvac–Phi Thermo Fischer Scientific Multilab 2000 spectrometer with an Al Kα radiation monochromator at 1486.6 eV) was carried out to confirm the composition and oxidation state of the prepared Sn/SnO_2_/C material.

### 2.3. Electrochemical Measurement

In order to investigate the electrochemical performance of the synthesized Sn/SnO_2_/C material, working electrodes were prepared by a slurry of 70% active materials, 20% carbon black (conductive agent), and 10% polyvinyl difluoride (PVDF) used as a binder in N-methy1-2-pyrrolidone (NMP). The mixture was then stirred with a magnetic stirrer for 2 h to obtain a homogenous solution. After that, the slurry was bladed on copper foil (current collector) and then transferred to a vacuum oven at 80 °C overnight. The prepared dry electrode was cut into a 12 mm round shape with an active mass of about 1.4 mg, and was pressed at 20 MPa. The coin cells (CR-2032) were assembled inside the glovebox (water and oxygen content < 1), where lithium metal foil was used as a reference and counter electrode. The propylene was used as a separator and 1M LiPF_6_ in EC:DEC = 1:1 (volume ratio) was used as an electrolyte. The galvanostatic charge/discharge curves were tested using a Land automatic batteries tester (LAND-CT2001A, Wuhan, China) in the potential range of 0–3 V as well at different current densities. The cyclic voltammetry (CV) was tested at a scan rate of 0.1 V at the electrochemical workstation (CHI660D, Shanghai, China). The electrochemical impedance spectroscopy (EIS) tests were also performed in the frequency ranging from 0.01 Hz to 1 MHz to check the ionic conductivity of the prepared electrodes.

## 3. Results

We first prepared a Sn/SnO_2_/C nano-composite by using the facile hydrothermal method along with high-temperature post-treatment of about 800 °C. In order to investigate the crystalline and phase confirmation of the prepared materials, X-ray diffraction Broker D8 with Cu Kalpha radiation was performed. Figure 1 shows the XRD pattern of the synthesized Sn/SnO_2_/C. The XRD pattern shows the prominent diffraction peaks of tin (JCPDS No 90-08570) with high intensity, where some of the diffraction peaks of the SnO_2_ phase structure were also observed and matched well with the SnO_2_(JCPDS No 14-1445), which had comparatively very low intensity. The diffraction peaks belonging to the SnO_2_ crystal structure can be observed in the XRD pattern. The observed diffraction peaks have very low intensity when compared with the prominent phase structure of tin. The X-ray diffraction pattern shows the existence of both structures in the composite, where the low intensity diffraction peaks suggest that the SnO_2_ phase shows their existence in the synthesized composite material in a low ratio, or the high-intensity peaks depressed their execution in the pattern.

To investigate and characterize the surface and morphological characteristics of the prepared Sn/SnO_2_/C nano-composite, scanning electron microscope (SEM) was performed and the results are presented in Figure 2. Figure 2a contains the typical low-magnification images of the Sn/SnO_2_/C nano-composite. It also shows particles irregular in shape as well as size, which may be due to the two different phases generated during the synthesis process. No obvious agglomerates were observed for the tin particles in the pattern. From the high-magnification images presented in Figure 2b, it can be seen clearly that the irregularity in terms of size and shape is due to the transformation of glucose and carboxymethyl cellulose (CMC) into carbon matrix during carbonization and the high-temperature (800 °C) treatment. The observed irregularity of the particles in the presented Figure 2a,b, suggests that the prepared Sn/SnO_2_/C nano-composites were inter-connected with each other, which created short pathways and more chances for the penetration of electrolytes and more Lithium-ion diffusion. Energy dispersive spectroscopy (EDS) along with SEM was performed to confirm the elemental mapping for the Sn/SnO_2_/C nano-composite and the results are shown in Figure 2c–f. It can be seen in the figure that tin agglomerates are distributed in the carbon matrix; the existence of oxygen mapping in Figure 2f can also be observed. No other contaminations were observed for other particles, which confirms that Sn and SnO_2_ particles were distributed in the carbon matrix and further clarifies the even distribution of the Sn in the carbon matrix.

Transmission electron microscope (TEM) analysis along with high-resolution transmission electron microscope (HRTEM) and selected area electron diffraction (SAED) analysis was performed to investigate and obtain more structural information of the as-prepared Sn/SnO_2_/C nano-composite anode, as shown in Figure 3. The presented TEM image in Figure 3a shows the even distribution of Sn and SnO_2_ nanoparticles in the carbon matrix. The SAED pattern in Figure 3b shows different lattice fringes, where in fact no obvious patterns can be differentiated easily for any specific phases, which further confirms and indicates the formation of a complex composite structure. A high-resolution TEM was used to reveal the lattice fringes of the prepared Sn/SnO_2_/C nano-composite as shown in Figure 3c–e. In the HRTEM image of Figure 3c, two different crystalline domains are embedded in the carbon matrix and are enlarged in the selected area of the image and are further presented in Figure 3d,e to confirm information about their lattices. In Figure 3d, the crystalline structure of Sn with its (200) planes can be identified. Figure 3e contains a SnO_2_ crystalline structure with its (110) planes. The inter-atomic spacing distance of 0.293 nm [29] and 0.334 nm [30] corresponding to (200) and (110) for both Sn and SnO_2_ match well with the XRD results. The observed lattice fringes of the SnO_2_ crystal structure in Figure 3e confirm their existence in the carbon matrix as well in the composite. The corresponding TEM and HRTEM images confirm the Sn/SnO_2_/C nano-composite phase and distinguish two crystalline structures embedded in the carbon matrix. As the literature reports, Sn itself as an anode suffers from high volume expansion, whereas SnO_2_ tends to agglomerate into large particles, both of which induce the fading of the performance of the battery. So, fabricating a composite structure has advantages such as controlling the volume expansion of Sn and depressing the agglomerates of the SnO_2_, which is expected to improve the reaction efficiency of the Sn/SnO_2_/C composite structure.

The chemical composition and oxidation states of the as-prepared Sn/SnO_2_/C nano-composite was further scrutinized via X-ray photo electron spectroscopy (XPS), as shown in Figure 4a–d. In the low-resolution survey spectra depicted in Figure 4a, the typical peaks of carbon, tin, and oxygen can be identified, indicating the purity of the synthesized Sn/SnO_2_/C nano-composite samples. As shown in Figure 4b, two strong characteristic peaks belonging to Sn3d_5/2_ and Sn3d_3/2_ of Sn/SnO_2_/C can be observed at 487.2 and 496.6 eV [31]. These characteristic bands of Sn/SnO_2_/C confirmed the oxidation of Sn into Sn^+4^ states [32]. Figure 4c represents the high-resolution spectra of Cs1 at 286.1 eV, which further confirm the existence of carbon in the composite sample. The major peak of carbon in the spectra indicates the existence of C species in the Sn/SnO_2_/C nano-composite. 

Figure 4d shows the high-resolution spectra of Os1 with maximum intensity at 533.6 eV attributed to the C–O bonding functional group, which further contributes to the reversibility of LiO_2_ during the cycling process [33].

In order to explore the electrochemical performance of the prepared Sn/SnO_2_/C nano-composite, half-coin cells were assembled in a glovebox where metallic Li foils were used as counter and reference electrodes, and the corresponding synthesized Sn/SnO_2_/C nano-composite materials were used as a negative electrode. The measured electrochemical performance of the Sn/SnO_2_/C nano-composite is depicted in Figure 5a–c. The cyclic voltammetry (CV) results were tested in the potential window ranging from 0.01 to 3 V (vs. Li^+^/Li) at a scanning rate of 0.1 mV s^−1^ for the initial five cycles for LIBs and are shown in Figure 5a. As observed in Figure 5a, in the first cycle there were six cathodic peaks appearing at different voltages and are identified at 0.338 V, 0.76 V, 1.5 V, 1.7 V, 1.8 V and 2.8 V, respectively. All the observed peaks in the first cycle disappeared, which was attributed to the solid electrolyte interphase (SEI) layer forming on the surface of the active electrodes. Moreover, all the observed oxidation and reduction peaks in the subsequent cycles are distinct and overlap each other, which indicates a stable electrochemical performance. The characteristic electrochemical reduction and oxidation peaks reflect the electrochemical behavior of the Sn and SnO_2_ anodes in one electrode system for LIBs.

The measured electrochemical impedance (EIS) of the Sn/SnO_2_/C nano-composite for LIBs is shown in Figure 5b. In general, a Nyquist plot contains a semicircle and a straight line representing charge transfer in high frequency and lower frequency regions, respectively. The depicted EIS results in Figure 5b show lower resistance with a high rate diffusion of Li-ion in the Sn/SnO_2_/C nano-composite’s negative electrode for LIBs. The inset in Figure 5b represents the EIS results measured for pure Sn as a negative electrode for LIBs, which show high conductivity in a low-frequency range. The absence of the high-frequency region may be due to the ESI layer formation/ decomposition in the pure Sn electrode system which further enhanced the diffusion rate of Li-ion in the Sn/SnO_2_/C nano-composite electrode for LIBs. Figure 5c represents the cycle performance of the Sn/SnO_2_/C nano-composite electrode and pure tin electrode for LIBs measured at 100 and 500 mAg^−1^ current densities in a potential range 0–3 V for 100 cycles. The first 5 cycles were measured at 100 mAg^−1^ and the remaining 90 cycles were measured at 500 mAg^−1^. The initial discharge capacity of the Sn/SnO_2_/C nano-composite was 2248 mAhg^−1^ with coulombic efficiency 70% of first charge/discharge capacity and a 99% capacity was maintained for the remaining cycles, even at high current density. In Figure 5c, we can also see the cycle performance of the pure tin electrode measured at same current densities in order to compare the cycle performance of both electrodes. Here, we can see the capacity drops after a few cycles at high current density; high capacity decay may be due to the high volume expansion or because of the low ionic conductivity of the tin electrode for LIBs. On the other hand, the discharge capacity decreased to 1685 mAhg^−1^ in the second cycle for the Sn/SnO_2_/C nano-composite electrode and then maintained up to 95% for the remaining cycles. The first high discharge capacity may be attributed to the SEI layer on the surface of the electrode which further decomposed in subsequent cycles as a result of a large amount of capacity decay. It is believed that such a high capacity with excellent coulumbic efficiency may be attributed to the combination of Sn and SnO_2_ in the conductive carbon matrix [34]. The Sn/SnO_2_/C nano-composite shows a high reversible capacity of about 489 mAhg^−1^ at the 100th cycle. As a result, the Sn/SnO_2_/C nano-composite showed enhanced electrochemical performance for LIBs and could be a promising candidate as a negative electrode for future prospects.

## 4. Conclusions

In this work, a Sn/SnO_2_/C nano-composite was synthesized via a hydrothermal method as an anode for LIBs. The basic structural characterization performed using SEM and TEM revealed the existence and homogenous distribution of the Sn nano-particles and SnO_2_ in the carbon matrix, which significantly enhanced the ionic conductivity of the electrode and buffered the volume expansion during repeated lithiation/delithiation processes. Moreover, the synthesized Sn/SnO_2_/C nano-composite showed high initial first-cycle discharge capacity (2248 mAhg^−1^) at 100 mAg^−1^ with a first-cycle coulombic efficiency of 70% and also displayed 489 mAhg^−1^ at a relatively high current density of about 500 mAg^−1^ after 100 cycles. As a result, high electrochemical active and crystalline nanoparticles embedded in the carbon matrix anode were achieved. The improved electrochemical performance of the prepared Sn/SnO_2_/C nano-composite enables it to be a promising anode for next-generation LIBs.

## Figures and Tables

**Figure 1 materials-15-02475-f001:**
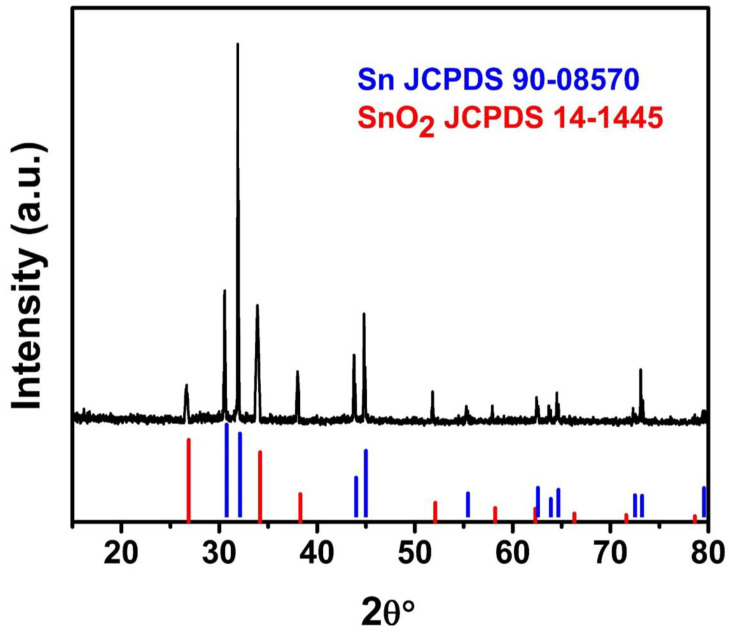
XRD pattern of Sn/SnO_2_/C.

**Figure 2 materials-15-02475-f002:**
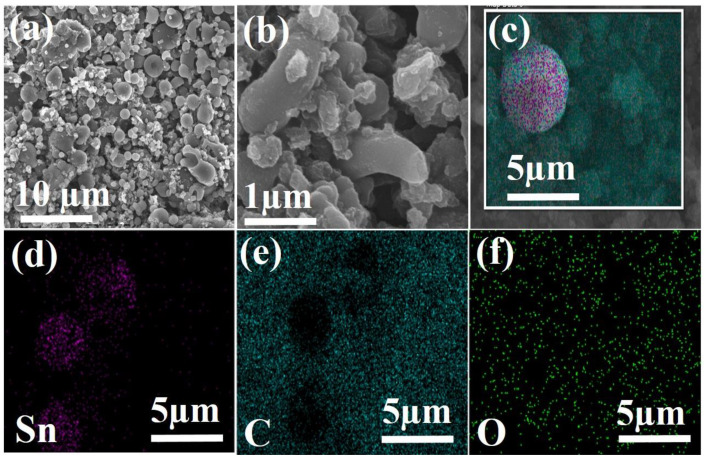
(**a**,**b**) SEM images; and (**c**–**f**) EDS and the corresponding elemental mapping Sn, C and O of as-prepared Sn/SnO_2_/C nano-composite.

**Figure 3 materials-15-02475-f003:**
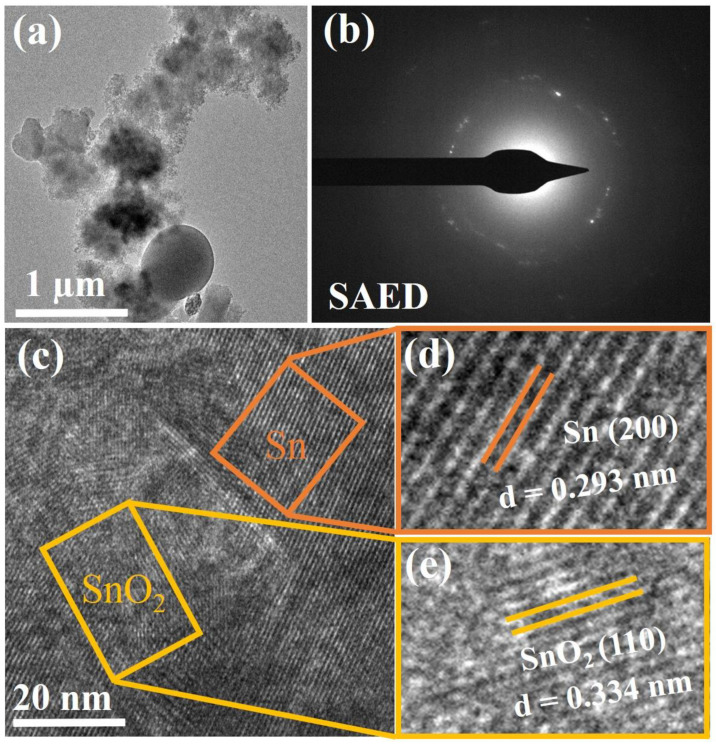
(**a**) TEM image of the Sn/C composite; (**b**) SAED pattern of the composite; and (**c**–**e**) HRTEM and enlarged image of the Sn/SnO_2_/C nano-composite.

**Figure 4 materials-15-02475-f004:**
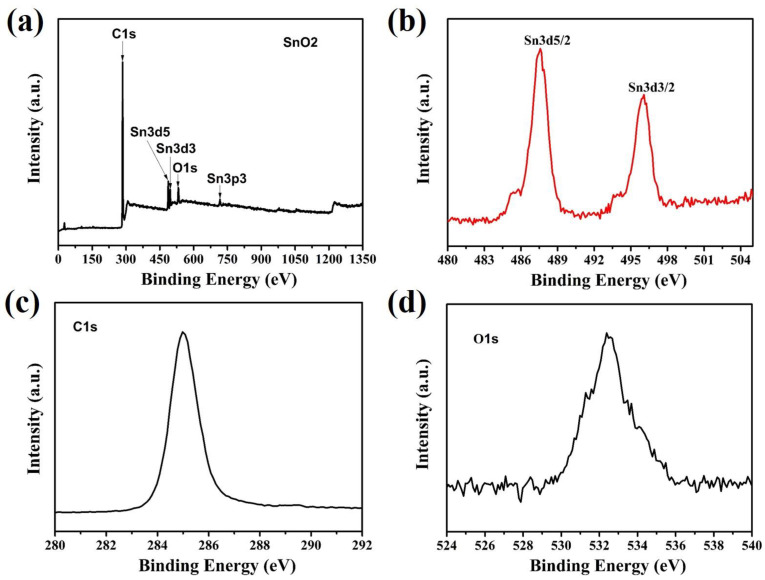
XPS spectra of Sn/SnO_2_/C: (**a**) survey scan; (**b**) Sn3d spectra; (**c**) C1s spectra; and (**d**) O1s spectra.

**Figure 5 materials-15-02475-f005:**
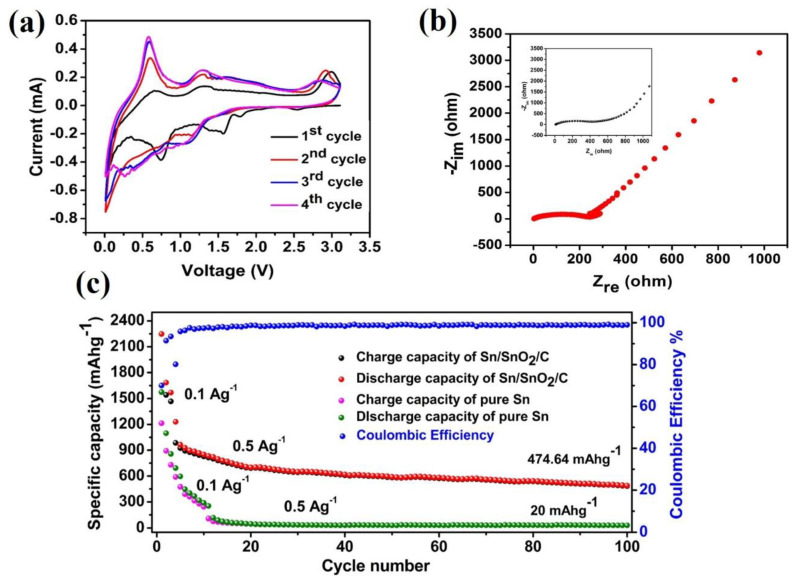
Electrochemical performance of Sn/SnO_2_/C nano-composite for LIBs: (**a**) initial five cyclic voltammetry curves scanned from 0.01 to 3 V at a rate of 0.01 mV s^−1^; (**b**) EIS of Sn/SnO_2_/C nano─composite; and (**c**) cycle performance of Sn/C electrode acquired at two different current densities of about 100 mA g^−1^ and 500 mAg^−1^.

## Data Availability

Data sharing is not applicable for this article.

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
