# Peer review of "Synthesis and Characterization of Sn/SnO2/C Nano-Composite Structure: High-Performance Negative Electrode for Lithium-Ion Batteries"

_materials, 2022, doi:10.3390/ma15072475_

Round 1
Reviewer 1 Report
Manuscript ID: materials-1592431
JOURNAL: MDPI Materials
TITLE: Synthesis and characterization of Sn/SnO2/C nano-composite structure: high-performance negative electrode for lithium-ion batteries.
The work basically describe the development and characterization of Sn/SnO2/C nano-composite via a hydrothermal approach and the electrochemical characterization of these developed nanocomposites. The presented work is interesting however there are serious concerns regarding the work which should be addressed, in case the editor feels the work is acceptable for publishing in Materials. To say some of them:
- Addressing the novelty of the work. Apparently, plenty of literatures are available on Tin/Tin oxide/carbon nanocomposite system and based materials as anodes for LIBs. If the authors claim the new hydrothermal processing is novel, then the authors should put some effort to make this clear to the readers. The process can be compared with other hydrothermal techniques used for processing similar composites.
- There are so many mistakes, typos, and wrongly used words: some examples (just some of them).
- “maintenance” (page 2, 83-84) ?
- “analytical grid”and page 3, 95
- of pistil and mortar and then annealed at 800 °C with the rate of 5C/min for page 3, 103
- 4hrs under argon atmosphere. Page 3 ,104
- crystalline and phase confirmation of the prepared materials, X-ray diffractom page 4, 141
- which was not observed much page5, 192
- Transmission electron microscope TEM along with high-resolution transmission electron microscope HRTEM and selected area electron diffraction SAED were performed to investigate more structural insight information of the prepared Sn/SnO2/C nano-composite material, and are presented in Fig. 3. (page 5, 176-179).
- “reaming cycles even at high current”. (Page 9, 275 )
- “are disctinct”. Page 8, 253
The authors should read the write-up couple of times at least and also should do an extensive check before uploading it again.
- From the X ray diffractogram presented the authors claim the presence of peaks corresponding to Sn and SnO2. Please mark the peaks corresponding to Sn and SnO2 on the diffractogram and attach the references in the text . It would be interesting to know what is JSPD NO?? I have heard of JCPDS files for XRD. Please clarify.
- SEM:
“Here, in Fig. 2(a,b) contains high-magnification images of the 157 Sn/SnO2/C nano-composite…………………….No agglomerates were observed obviously for the tin particles in the observed pattern. As 160 the low magnification images are presented in Fig 2c”
Are you sure about the magnification??
- SEM-EDS : Please mention/show clearly the image of which the mapping is presented. Its confusing.
- “Figure 2. (a-c) SEM images (d) EDS and the corresponding elemental mapping (e-f) of Sn/SnO2/C 174 nano-composite”. The figure title is just confusing!
- TEM: please provide some reference to the “d spacing” reported.
The work need extensive English check and clarity in discussing the material preparation part.
Author Response
Comments and Suggestions for Authors
Manuscript ID: materials-1592431
JOURNAL: MDPI Materials
TITLE: Synthesis and characterization of Sn/SnO2/C nano-composite structure: high-performance negative electrode for lithium-ion batteries.
The work basically describe the development and characterization of Sn/SnO2/C nano-composite via a hydrothermal approach and the electrochemical characterization of these developed nanocomposites. The presented work is interesting however there are serious concerns regarding the work which should be addressed, in case the editor feels the work is acceptable for publishing in Materials. To say some of them:
- Addressing the novelty of the work. Apparently, plenty of literatures are available on Tin/Tin oxide/carbon nanocomposite system and based materials as anodes for LIBs. If the authors claim the new hydrothermal processing is novel, then the authors should put some effort to make this clear to the readers. The process can be compared with other hydrothermal techniques used for processing similar composites.
- There are so many mistakes, typos, and wrongly used words: some examples (just some of them).
- “maintenance” (page 2, 83-84) ?
- “analytical grid”and page 3, 95
- of pistil and mortar and then annealed at 800 °C with the rate of 5C/min for page 3, 103
- 4hrs under argon atmosphere. Page 3 ,104
- crystalline and phase confirmation of the prepared materials, X-ray diffractom page 4, 141
- which was not observed much page5, 192
- Transmission electron microscope TEM along with high-resolution transmission electron microscope HRTEM and selected area electron diffraction SAED were performed to investigate more structural insight information of the prepared Sn/SnO2/C nano-composite material, and are presented in Fig. 3. (page 5, 176-179).
- “reaming cycles even at high current”. (Page 9, 275 )
- “are disctinct”. Page 8, 253
The authors should read the write-up couple of times at least and also should do an extensive check before uploading it again.
- From the X ray diffractogram presented the authors claim the presence of peaks corresponding to Sn and SnO2. Please mark the peaks corresponding to Sn and SnO2 on the diffractogram and attach the references in the text . It would be interesting to know what is JSPD NO?? I have heard of JCPDS files for XRD. Please clarify.
- SEM:
“Here, in Fig. 2(a,b) contains high-magnification images of the 157 Sn/SnO2/C nano-composite…………………….No agglomerates were observed obviously for the tin particles in the observed pattern. As 160 the low magnification images are presented in Fig 2c”
Are you sure about the magnification??
- SEM-EDS : Please mention/show clearly the image of which the mapping is presented. Its confusing.
- “Figure 2. (a-c) SEM images (d) EDS and the corresponding elemental mapping (e-f) of Sn/SnO2/C 174 nano-composite”. The figure title is just confusing!
- TEM: please provide some reference to the “d spacing” reported.
The work need extensive English check and clarity in discussing the material preparation part.

Reviewer 2 Report
In this paper the authors reported an easy method to prepare Sn/SnO2/C composite, and its excellent electrochemical performance was shown. This paper can published after revision.
1. There are two paragraphs in p. 5 and p. 6 explaining Fig. 3. The two paragraphs should be combined.
2. Line 185, “Fig. 3(b) may be “Fig. 3(c)”
3. line 252, “solid electrolyte interface” may be “solid electrolyte interphase”.
4. There are many typographic errors. The manuscript should be revised by a professional editor.
The coulombic efficiency in Fig. 5c is excellent.
So the data is acceptable for publication.
Additional revision.
In the impedance data in Fig. 5b, it is better to
adjust the scales of x-axis and y-axis in order to
see the degree of deviation of high frequency data from a circle.
Author Response
Comments and Suggestions for Authors
In this paper the authors reported an easy method to prepare Sn/SnO2/C composite, and its excellent electrochemical performance was shown. This paper can published after revision.
- There are two paragraphs in p. 5 and p. 6 explaining Fig. 3. The two paragraphs should be combined.
- Line 185, “Fig. 3(b) may be “Fig. 3(c)”
- line 252, “solid electrolyte interface” may be “solid electrolyte interphase”.
- There are many typographic errors. The manuscript should be revised by a professional editor.
The coulombic efficiency in Fig. 5c is excellent.
So the data is acceptable for publication.
Additional revision.
In the impedance data in Fig. 5b, it is better to
adjust the scales of x-axis and y-axis in order to
see the degree of deviation of high frequency data from a circle.

Reviewer 3 Report
Title:
Synthesis and characterization of Sn/SnO2/C nano-composite structure: high-performance negative electrode for lithium-ion batteries.
Comments:
The authors reported a hydrothermal synthesis of Sn/SnO2/C nano-composite materials to apply as anode in lithium-ion batteries. The searching for new active materials is important due to the necessity innovation in the battery field and associated with environmental approach is very timeless and opportune. Although that, the work present drawbacks as:
Point 1: Active mass loading used should be mentioned in the work.
Point 2: In Figure 2 d-f what is the scale?
Point 3: In line 167 we can find: “Energy dispersive spectroscopy EDS along with SEM was performed to confirm the elemental mapping for Sn/SnO2/C nano-composite and are shown in Fig. 2(d), e and f. It can be seen in the Fig that tin particles are uniformly distributed in the carbon matrix and we can also see the existence of oxygen mapping in Fig. 2(f).” Analaizing the EDS image Fig 2d we can find an agglomeration of Sn atom. This should be mentioned and the text must be change.
Point 4: In Fig 5c the cycle performance of pure Sn should be added to demonstrate the differences between both samples.
Point 5: Authors demonstrate that this sample shows high discharge capacity (at abstract and conclusion) of 2248 mAhg-1 but this value was obtained only for the first cycle, before the SEI formation. More focus should be pointed to the discharge capacity of 489 mAhg-1. This information should be re-written. Also the reference as (http://dx.doi.org/10.1016/j.electacta.2015.10.185) should be added to compare the morphology and performance.
According to that, this referee believes that this work should considered to major revision.
Author Response
Comments and Suggestions for Authors
Title:
Synthesis and characterization of Sn/SnO2/C nano-composite structure: high-performance negative electrode for lithium-ion batteries.
Comments:
The authors reported a hydrothermal synthesis of Sn/SnO2/C nano-composite materials to apply as anode in lithium-ion batteries. The searching for new active materials is important due to the necessity innovation in the battery field and associated with environmental approach is very timeless and opportune. Although that, the work present drawbacks as:
Point 1: Active mass loading used should be mentioned in the work.
Point 2: In Figure 2 d-f what is the scale?
Point 3: In line 167 we can find: “Energy dispersive spectroscopy EDS along with SEM was performed to confirm the elemental mapping for Sn/SnO2/C nano-composite and are shown in Fig. 2(d), e and f. It can be seen in the Fig that tin particles are uniformly distributed in the carbon matrix and we can also see the existence of oxygen mapping in Fig. 2(f).” Analaizing the EDS image Fig 2d we can find an agglomeration of Sn atom. This should be mentioned and the text must be change.
Point 4: In Fig 5c the cycle performance of pure Sn should be added to demonstrate the differences between both samples.
Point 5: Authors demonstrate that this sample shows high discharge capacity (at abstract and conclusion) of 2248 mAhg-1 but this value was obtained only for the first cycle, before the SEI formation. More focus should be pointed to the discharge capacity of 489 mAhg-1. This information should be re-written. Also the reference as (http://dx.doi.org/10.1016/j.electacta.2015.10.185) should be added to compare the morphology and performance.
According to that, this referee believes that this work should considered to major revision.

Round 2
Reviewer 1 Report
Manuscript ID: materials-1592431_R2
JOURNAL: MDPI Materials
TITLE: Synthesis and characterization of Sn/SnO2/C nano-composite structure: high-performance negative electrode for lithium-ion batteries.
The reviewers have addressed many of the queries raised. However, some more observations are there as listed below. The manuscript can be accepted after those minor corrections.
- “abserved” should be observed. (page 5, line 173). Typos even at this level is bad and also extensive English check would be required (I hope which can be done at the manuscript editing stage).But please check.
- Figure 2: if the elemental mapping of a particular SEM image is presented, kindly include the same image on the main manuscript (preferably) of at least in the secondary information. If the image is given kindly say the “ elemental mapping of this image” on the figure label.
- The inter-atomic spacing distance 0.293 nm and 0.334 nm corresponding to (-101) and (110) for both Sn and SnO2 are matching well with the XRD (page 193-194).
Did you find any literatures for similar d-spacing for Sn and SnO2?
It would be good to add.

Author Response
The reviewers have addressed many of the queries raised. However, some more observations are there as listed below. The manuscript can be accepted after those minor corrections.
- “abserved” should be observed. (page 5, line 173). Typos even at this level is bad and also extensive English check would be required (I hope which can be done at the manuscript editing stage).But please check.
- Figure 2: if the elemental mapping of a particular SEM image is presented, kindly include the same image on the main manuscript (preferably) of at least in the secondary information. If the image is given kindly say the “ elemental mapping of this image” on the figure label.
- The inter-atomic spacing distance 0.293 nm and 0.334 nm corresponding to (-101) and (110) for both Sn and SnO2 are matching well with the XRD (page 193-194).
Did you find any literatures for similar d-spacing for Sn and SnO2?
It would be good to add.

Reviewer 3 Report
The reviewer agree with the changes made on the manuscript.